# Sensitization of meningeal afferents to locomotion-related meningeal deformations in a migraine model

**Andrew S Blaeser[1], Jun Zhao[1], Arthur U Sugden[2], Simone Carneiro-Nascimento[1], Mark L Andermann[2,3]\*, Dan Levy[1]\***

[1]Department of Anesthesia, Critical Care and Pain Medicine, Beth Israel Deaconess Medical Center, Harvard Medical School, Boston, United States; [2]Division of Endocrinology, Diabetes, and Metabolism, Department of Medicine, Beth Israel Deaconess Medical Center, Harvard Medical School, Boston, United States; [3]Department of Neurobiology, Harvard Medical School, Boston, United States

**\*For correspondence:**
manderma@bidmc.harvard.edu
(MLA);
dlevy1@bidmc.harvard.edu (DL)

**Abstract** Migraine headache is hypothesized to involve the activation and sensitization of trigeminal sensory afferents that innervate the cranial meninges. To better understand migraine pathophysiology and improve clinical translation, we used two-photon calcium imaging via a closed cranial window in awake mice to investigate changes in the responses of meningeal afferent fibers using a preclinical model of migraine involving cortical spreading depolarization (CSD). A single CSD episode caused a seconds-long wave of calcium activation that propagated across afferents and along the length of individual afferents. Surprisingly, unlike previous studies in anesthetized animals with exposed meninges, only a very small afferent population was persistently activated in our awake mouse preparation, questioning the relevance of this neuronal response to the onset of migraine pain. In contrast, we identified a larger subset of meningeal afferents that developed augmented responses to acute three-dimensional meningeal deformations that occur in response to locomotion bouts. We observed increased responsiveness in a subset of afferents that were already somewhat sensitive to meningeal deformation before CSD. Furthermore, another subset of previously insensitive afferents also became sensitive to meningeal deformation following CSD. Our data provides new insights into the mechanisms underlying migraine, including the emergence of enhanced meningeal afferent responses to movement-related meningeal deformations as a potential neural substrate underlying the worsening of migraine headache during physical activity.

## eLife assessment

This **fundamental** study explored the impact of migraine-related cortical spreading depression (CSD) on the firing of nerves innervating the coverings of the brain that are considered the putative source of migraine-related pain. Using **convincing** approaches they show that these responses are altered in response to mechanical deformation of the brain coverings. Given that migraine is characterized by worsening head pain in response to movement, the findings offer a potential mechanism that may explain this clinical phenomenon.

## Introduction

A large body of evidence supports the notion that migraine headache involves the trigeminal meningeal sensory system (*Ashina et al., 2019*; *Levy and Moskowitz, 2023*). Persistent discharge of meningeal afferents is thought to mediate the ongoing headache, while their augmented mechanosensitivity

has been suggested to underlie migraine headache exacerbation during normally innocuous physical activities that cause transient intracranial hypertension, such as coughing and other types of straining (*Blau and Dexter, 1981*). Current understanding of migraine-related responses of meningeal afferents is largely based on animal models. For example, triggering an episode of cortical spreading depolarization (CSD), a self-propagating wave of neuronal and glial depolarizations thought to mediate migraine aura, causes persistent activation and mechanical sensitization of meningeal afferents (*Zhang et al., 2010*; *Zhao and Levy, 2015*; *Zhao and Levy, 2016*).

Despite the preclinical evidence implicating enhanced responsiveness of meningeal afferents as a driver of migraine headache (*Levy and Moskowitz, 2023*), these studies have almost all used acute invasive experiments involving electrophysiological recordings in anesthetized animals with surgically exposed and mildly inflamed meninges (*Levy et al., 2007*). Moreover, studies documenting the mechanical sensitization of meningeal afferents were based on findings of increased responsiveness to artificial compressive forces applied to the meninges of a depressurized brain. Hence, there is a significant gap in our understanding of whether and how meningeal afferents respond to migraine-related events under more naturalistic conditions in behaving animals with an intact and pressurized intracranial space.

To better understand migraine pathophysiology and improve clinical translation, we leveraged a newly developed approach for two-photon calcium imaging of meningeal afferent responses within the closed intracranial space of an awake-behaving mouse (*Blaeser et al., 2022d*) in the CSD model of migraine. We studied changes in afferent ongoing activity and afferent responses to three-dimensional (3D) meningeal deformation associated with locomotion *Blaeser et al., 2022d* following the triggering of a single CSD episode. Our data provides new insights into the mechanisms underlying migraine pathophysiology, including acute calcium signaling in meningeal afferent fibers as a potentially critical nociceptive factor contributing to migraine pain and the emergence of enhanced meningeal afferent responses to movement-related meningeal deformations as the neural substrate underlying the worsening of migraine headache during physical activity.

## Results

### Propagating calcium activity across afferent fibers during CSD

To investigate meningeal afferent responses to CSD, we performed two-photon calcium imaging of GCaMP6s-expressing trigeminal afferent fibers innervating the meninges above the visual cortex (n=325 fibers from 9 fields of view [FOVs] from 7 mice, *Figure 1A*). We triggered a single CSD episode in the frontal cortex with a cortical pinprick. In every experiment (9 CSDs in 7 mice), we detected a slow, CSD-like wave of calcium activity in numerous meningeal afferent fibers within 1 min following the pinprick (*Figure 1B*, *Figure 1—video 1*) as well as in background regions (likely reflecting signal from small, out-of-focus afferent branches). These calcium waves proceeded from the pinprick site in an anterior-to-posterior direction across the FOV (*Figure 1C*). We also observed progressive activation of portions of individual afferent fibers aligned to the wave's movement direction. To characterize this phenomenon, we focused on sets of regions of interest (ROIs) belonging to the same long afferent fiber oriented along the direction of the calcium wave (*Figure 1D*, *Figure 1—video 1*). Compared to baseline afferent calcium signals observed during periods of locomotion, during which all ROIs belonging to an afferent were activated near-simultaneously, as previously reported (*Blaeser et al., 2022d*), the sequential recruitment of ROIs along an afferent fiber during the CSD-like wave was much slower (*Figure 1E–G*). The proportion of afferents activated during this period exceeded the proportion activated during locomotion bouts (*Figure 1I*). The magnitude of activation was also larger (*Figure 1J*).

### Acute afferent activation is not related to CSD-evoked meningeal deformation

CSD gives rise to acute neuronal and glial swelling and shrinkage of the cortical extracellular space (*Mazel et al., 2002*; *Takano et al., 2007*; *Rosic et al., 2019*). Such cortical mechanical perturbations could lead to acute meningeal deformation, which we have shown previously can activate mechanosensitive meningeal afferents (*Blaeser et al., 2022d*). We postulated that if CSD leads to meningeal deformations, these deformations could drive the acute afferent response during the

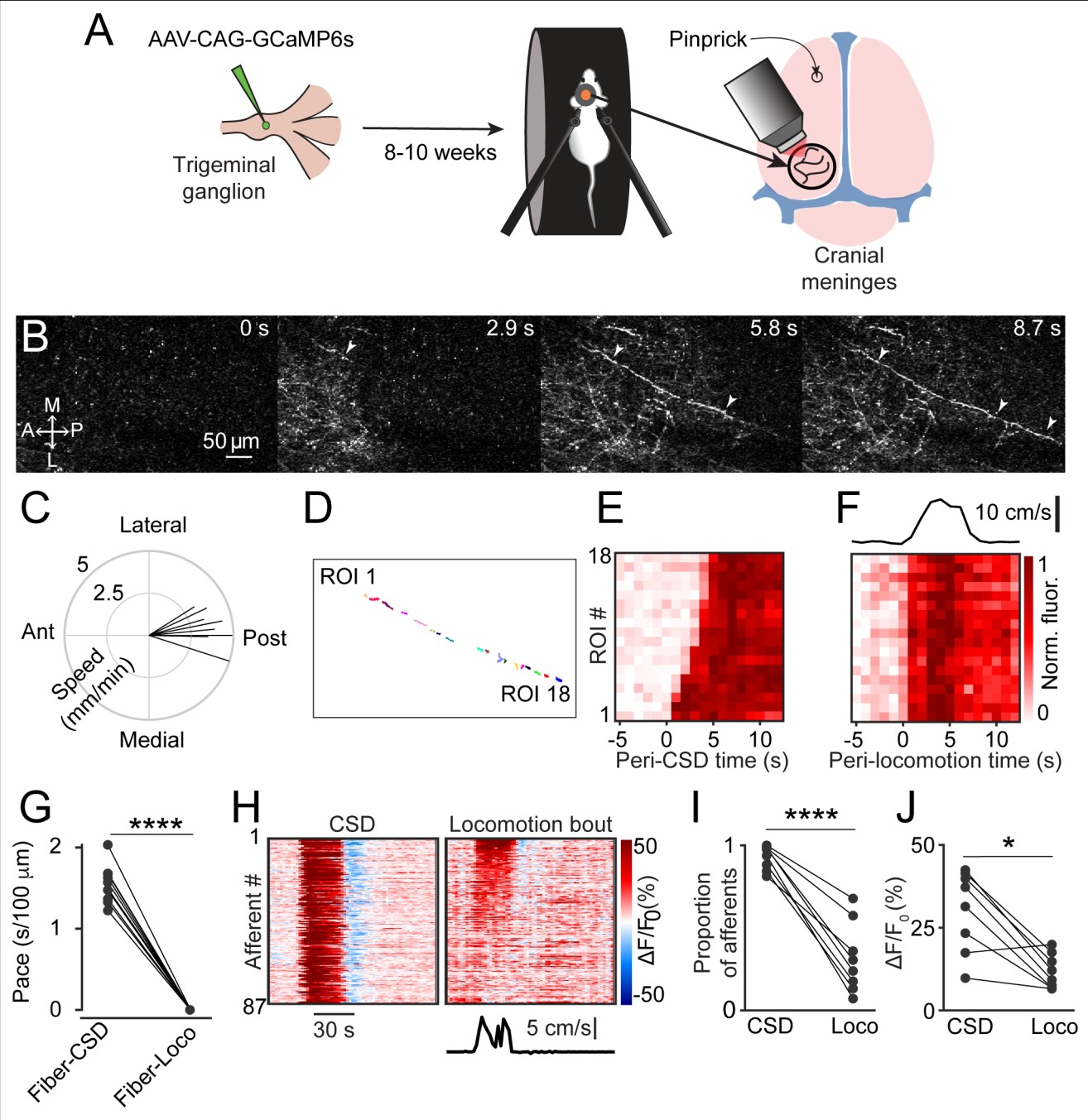

**Figure 1.** Cortical spreading depolarization (CSD) drives wave-like calcium activity in meningeal afferents. (**A**) Mice received a trigeminal ganglion injection of an AAV to express GCaMP6s in trigeminal meningeal afferents. After 8–10 weeks, following the implantation of a headpost and a cranial window, mice were habituated to head restraint and subjected to two-photon calcium imaging while head-fixed on a running wheel to study the effect of pinprick-triggered CSD on the activity of meningeal afferents. (**B**) Example of a CSD-associated meningeal calcium wave that spreads across the field of view (FOV), with local segments of long afferent fibers becoming sequentially activated as the wave progresses (arrowheads). M: medial, L: lateral, A: anterior, P: posterior. (**C**) Summary of speed and direction of CSD-associated meningeal calcium waves, typically from anterior ('Ant.') (closer to where CSD was triggered anterior to the cranial window) to more posterior locations ('Post.'). Speed estimates were obtained using the analysis method described in *Figure 1—figure supplement 1*. On average, the wave progressed at 3.8±0.2 mm/min. (**D**) Map of 18 regions of interest (ROIs) belonging to a single meningeal afferent fiber visible in B. (**E**) Activity heatmap of the afferent ROIs indicated in D illustrating progressive activation in response to CSD. (**F**) In contrast, the same afferent ROIs became activated simultaneously during a locomotion bout. Top trace depicts locomotion speed. (**G**) The pace of the CSD-associated afferent calcium wave was much slower than the spread of activity along the same afferent fibers during locomotion-evoked activity pre-CSD (****p<0.0001, paired, two-tailed t-test). (**H**) Example heatmaps of afferent activity observed during CSD showing different time course

*Figure 1 continued on next page*

*Figure 1 continued*

and magnitudes when compared to the activity observed during a locomotion bout. Bottom trace depicts locomotion speed. (**I**) Comparisons across all FOVs indicate a higher proportion of afferents exhibiting acute activation during the CSD vs. during locomotion (****p<0.0001, iterated bootstrap). (**J**) A higher proportion of afferents also displayed increased magnitudes of activation (*p<0.05, paired, two-tailed t-test). See also *Figure 1—video 1*.

The online version of this article includes the following video and figure supplement(s) for figure 1:

**Figure supplement 1.** Further analyses of hemodynamics and calcium activity related to the cortical spreading depolarization (CSD) wave.

**Figure supplement 2.** Cortical spreading depolarization (CSD)-associated wave deformation and acute afferent responses.

**Figure 1—video 1.** Cortical spreading depolarization (CSD)-associated afferent calcium wave.

https://elifesciences.org/articles/91871/figures#fig1video1

Related to *Figure 1* and *Figure 1—figure supplement 1*. CSD-associated calcium waves spread from anterior to posterior, as shown for two example waves from different mice. Note the spread of calcium activation along individual afferents in each movie. Scale bars: 50 µm.

CSD wave. Assessment of meningeal deformation parameters (see Materials and methods) revealed severe meningeal scaling and shearing during the CSD-evoked afferent calcium wave (6 CSDs in 6 mice, *Figure 1—figure supplement 2A–D*). In some experiments, the temporal pattern of meningeal shearing (*Figure 1—figure supplement 2C*) somewhat resembled that of the acute afferent response. However, the pattern of meningeal scaling (*Figure 1—figure supplement 2B*) was incongruent with the acute afferent response. Surprisingly, the direction of Z-shifts in the meninges during this epoch was inconsistent across mice, resulting in no significant Z-shift on average relative to the pre-CSD epoch (*Figure 1—figure supplement 2D*). To estimate the relative contribution of the CSD-driven meningeal deformation to the acute afferent responses we observed, we used a general linear model (GLM; see *Blaeser et al., 2022d*, and Materials and methods). We focused on afferents whose pre-CSD (baseline) activity could be predicted by the GLM based on deformation predictors (n=145 afferents from 6 mice). We then plugged the peri-CSD deformation data into these GLMs to generate predictions of peri-CSD afferent activity and compared them to the observed activity. Overall, we observed a poor match between the real peri-CSD calcium signals and those predicted by the GLMs trained using pre-CSD deformation data (see example in *Figure 1—figure supplement 2E* and summary GLM fits in *Figure 1—figure supplement 2F*), suggesting that the model poorly predicted the magnitude of the activity and/or its temporal pattern in response to CSD. Because ~95% of the afferents were acutely activated by CSD (*Figure 1I*), we propose that this response is mostly driven by other non-mechanical factors, such as the local depolarizing effects of diffusible excitatory molecules.

## A minority of afferents exhibit prolonged activation or suppression after CSD

In anesthetized rats with exposed meninges, CSD drives sustained increases in ongoing activity lasting tens of minutes in ~50% of meningeal afferents (*Zhao and Levy, 2015*). To directly assess CSD-related changes in afferent ongoing activity in awake mice with intact meninges, we focused on afferent responses during epochs of immobility between locomotion bouts (8 CSDs in 7 mice). We observed low levels of ongoing activity at baseline before CSD (fluorescent events occurring 6.9 ± 0.3% of the time), consistent with our previous study in naïve mice (*Blaeser et al., 2022d*). Unexpectedly, most afferents (~70%, 201/288) did not display any change in ongoing activity during the 2 hr following CSD (termed 'post-CSD'). However, we identified sustained increases in ongoing activity in ~10% (30/288) of the afferents during this period. Surprisingly, we also observed a larger afferent population (~20%; 57/288) whose activity was suppressed (*Figure 2A–C*). Afferents with sustained activation showed increased ongoing activity that emerged at an ~25 min delay on average (*Figure 2D*). In contrast, afferents with sustained suppression showed decreases in ongoing activity beginning shortly after the passage of the acute calcium wave (*Figure 2D*). The durations of the afferent activation or suppression were similar, lasting ~25 min on average (*Figure 2E*).

## CSD augments afferent responsiveness associated with meningeal deformations

Meningeal deformation associated with locomotion bouts can lead to the activation of mechanosensitive meningeal afferents (*Blaeser et al., 2022d*). We wondered whether, following CSD, afferent responses to a given level of mechanical deformation would be enhanced (i.e. mechanical sensitization).

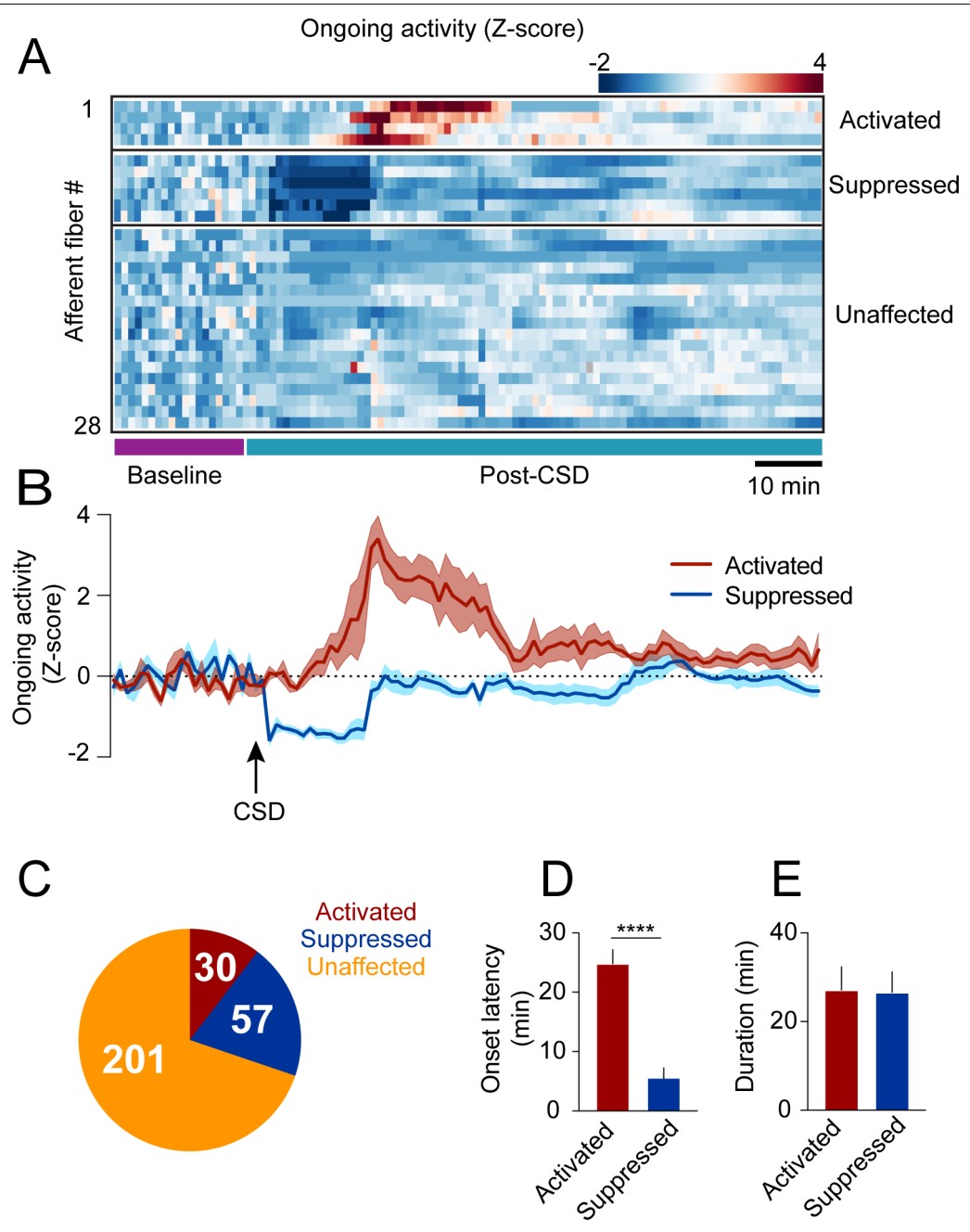

**Figure 2.** Cortical spreading depolarization (CSD)-related persistent changes in the ongoing activity of meningeal afferents. (**A**) Example heatmap of normalized ongoing activity (fraction of time afferents exhibited calcium events when the mouse is not locomoting) for all afferent fibers from a single field of view (FOV) during baseline and up to 120 min following CSD (termed '"post-CSD'). Data shows concatenated 1- min bins of activity. Afferents were either activated, suppressed, or unaffected by CSD. Note the delayed activation and immediate suppression in two small subsets of fibers. (**B**) Mean activity time course of the activated and suppressed afferents from the same population depicted in A. (**C**) Pie chart depicting the breakdown of the afferent subpopulations based on their change in ongoing activity following CSD. Most afferents were not affected (orange), while two smaller populations either exhibited prolonged activation (maroon) or suppression (blue) of ongoing activity following CSD (8 CSDs in 7 mice). (**D**) Afferents exhibiting prolonged activation had a longer onset latency than those exhibiting suppression (****p<0.0001, Mann-Whitney U-test. Error bars: SEM). (**E**) The duration of increases in ongoing activity and suppressions in activity were similar (p=0.97, two-tailed t-test. Error bars: SEM).

If so, this could explain the exacerbation of migraine headaches during physical activity. CSD suppresses cortical activity, leading to decreased motor function (*Houben et al., 2017*), including reduced locomotion in head-fixed mice (*Enger et al., 2017*). CSD-related vascular changes and reduced extracellular space (*Mazel et al., 2002*; *Takano et al., 2007*) could also affect meningeal deformations and the associated afferent response. Hence, we first analyzed the effect of CSD on wheel running activity and the associated meningeal deformation. In most sessions, mice stopped locomoting following the CSD (8/9 CSDs in 7 mice) but resumed sporadic wheel running activity ~6 min later on average (range 0.5–16.5 min). However, the mice ran less during the 2 hr following CSD than during the baseline period (*Figure 3B*). Locomotion bout analysis also revealed an overall reduction in bout rate during the post-CSD period (*Figure 3C*). Remarkably, despite the reduction in locomotion following CSD, we observed similar bout characteristics at baseline and post-CSD, including bout duration (*Figure 3D*) and peak velocity (*Figure 3E*). Given that CSD had minimal effect on locomotion bout characteristics, we next examined its effect on meningeal deformations. Surprisingly, CSD did not affect bout-related meningeal deformations: we observed similar scaling, shearing, and Z-shift values during the 2 hr post-CSD (*Figure 3F–H*).

Having shown that locomotion bout characteristics and the related meningeal deformations are not altered during the 2 hrs following CSD, we next compared afferent responses during locomotion bouts before and after CSD. Initial observations of afferent activation during locomotion indicated enhanced responsiveness following CSD (*Figure 4A*). To systematically investigate this augmented afferent response, we used GLMs (see Materials and methods and *Blaeser et al., 2022d*) to assess whether meningeal afferent activity becomes sensitized to the state of locomotion and/or to various aspects of meningeal deformation following CSD. We modeled each afferent's activity based on variables that describe (1) the binary state of locomotion, (2) mouse velocity, or (3) aspects of meningeal deformation, including scaling, shearing, and Z-shift.

We first focused on afferents whose activity could be predicted by the same variables both at baseline and following CSD (i.e. afferents that exhibited sensitivities to locomotion and/or deformation signals both before and after CSD, n=67/325 afferents, 9 CSDs in 7 mice). Higher GLM coefficients for a given variable post-CSD indicate greater afferent response during an equal expression level of that variable. Thus, we defined an afferent as sensitized by CSD if its GLM coefficients post-CSD were larger than at baseline (i.e. stronger activation of afferents per unit deformation or locomotion; for example, see *Figure 4C*). Using these criteria, we identified elevated locomotion and/or deformation-related activity (i.e. sensitization) post-CSD in ~51% of afferents (34/67; *Figure 4D*). In contrast, only 12% of afferents (8/67) showed reduced locomotion- and deformation-related activity (i.e. desensitization) post-CSD. Sensitivity was unchanged in the remaining 37% of afferents (25/67).

Meningeal afferent sensitization following CSD may reflect increased sensitivity to mechanical deformation and/or to other physiological processes that occur in response to locomotion (*Blaeser et al., 2022d*). Because locomotion and meningeal deformations are partially correlated (*Blaeser et al., 2022d*), we next estimated their relative contributions to the augmented afferent responsiveness post-CSD by comparing, for each sensitized afferent, the GLM coefficients generated for baseline epochs and for post-CSD epochs. Surprisingly, we found that only the deformation coefficients were increased post-CSD (*Figure 4E*), suggesting that meningeal afferent sensitization following CSD reflects primarily an increased sensitivity to local mechanical deformations.

We next considered the possibility that sensitization is also manifested in the unmasking of responsiveness to locomotion and/or meningeal deformation in previously silent (i.e. insensitive) meningeal afferents (*Levy and Strassman, 2002*; *Levy and Moskowitz, 2023*). Indeed, among the 245 afferents that were not well fit before CSD (i.e. insensitive), we detected a substantial population that developed a sensitivity to locomotion and deformation variables following CSD (53/245; i.e. afferents whose activity could be well predicted by locomotion and deformation variables following CSD). By contrast, far fewer neurons (13/245) lost their sensitivity to these variables after CSD (i.e. afferents that were well fit before but not after CSD). Overall, the above findings show that four times more afferents displayed increased sensitivity than decreased sensitivity following CSD (87 vs 21 afferents). This sensitized afferent population also displayed post-CSD increases in GLM coefficients related to deformation but not to locomotion (*Figure 4F*).

To further quantify the importance of the locomotion and deformation variables to the afferent sensitization, we estimated their relative contributions to the overall ability of the models to predict

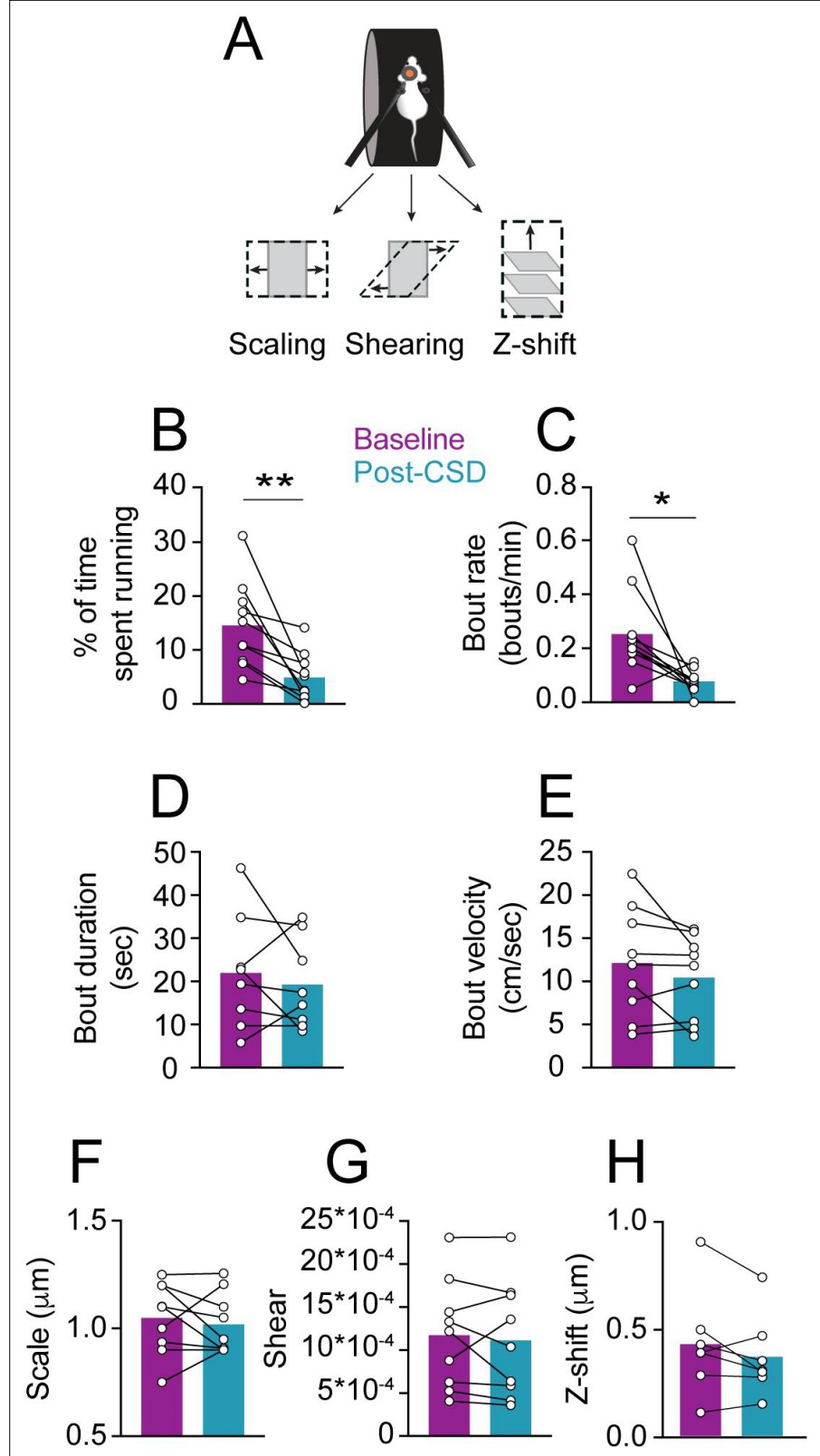

**Figure 3.** Locomotion and related meningeal deformations pre- and post-cortical spreading depolarization (CSD). (**A**) In head-fixed mice, wheel running is associated with meningeal scaling, shearing, and positive Z-shift (i.e. meningeal movement toward the skull). (**B**) When compared to the baseline period, there was an overall reduction in the time mice spent running during the 2 hr post-CSD observation period (**p<0.01, paired t-test, 9 CSDs in

*Figure 3 continued on next page*

*Figure 3 continued*

7 mice). (**C**) CSD also decreased locomotion bout rate (*p<0.05, Wilcoxon, signed rank test). (**D, E**) However, CSD did not affect bout duration (p=0.50, paired t-test) or bout peak velocity (p=0.18, paired t-test). (**F, G, H**) CSD also did not affect subsequent locomotion-evoked meningeal scaling, shearing, or Z-shift (p=0.56; p=0.55, p=0.18, paired t-tests, respectively, 9 CSDs, in 7 mice for scale and shear, 7 CSDs in 7 mice for Z-shift). Bars depict the mean.

the afferent activity pre- vs. post-CSD. To this end, we calculated the difference in model fit using the full model with all variables or models lacking either the set of deformation variables or locomotion variables. We found that the impact of deformation variables on the model fit was greater post-CSD than pre-CSD, while the impact of locomotion variables was similar pre- and post-CSD. This was true for afferents with models that were well fit both at baseline and following CSD (*Figure 4G*; purple subset in *Figure 4D*) and for those with models that were well fit only post-CSD (*Figure 4H*; magenta subset in *Figure 4D*). Further analysis revealed that scale, shear, and Z-shift deformations were, on average, equally important in predicting the activity patterns of sensitized afferents (*Figure 4I*). Taken together, these data suggest that the afferent sensitization following CSD primarily involves increased afferent responsiveness to a mix of meningeal deformation variables rather than to other locomotion-associated processes.

Previous studies in anesthetized rats suggested that the mechanisms underlying meningeal afferent mechanical sensitization are independent of those responsible for increased ongoing discharge in several migraine models, including CSD (*Levy and Strassman, 2002*; *Zhang et al., 2011*; *Zhang et al., 2013*; *Zhao and Levy, 2018*; *Zhao et al., 2021*). Here, using the CSD model in awake mice, we also found no association between sensitization and sustained changes in ongoing activity, as similar proportions of sensitized and non-sensitized afferents were activated, suppressed, or did not display any change in their ongoing activity following CSD (*Figure 4J*).

## Discussion

Prior studies suggested that CSD drives meningeal nociception that can lead to the headache phase in migraine with aura. These studies, which mostly involved invasive experiments in anesthetized rats with surgically exposed meninges, showed prolonged activation and mechanical sensitization of meningeal afferents (*Carneiro-Nascimento and Levy, 2022*). To better understand migraine pathophysiology and improve clinical translation, we used two-photon calcium imaging to characterize, for the first time, CSD-related changes in the responsiveness of individual meningeal sensory afferents at the level of their peripheral nerve fibers in the closed cranium of a behaving mouse. We show that a single CSD episode drives a wave of calcium activity across most afferents while producing a more prolonged change in ongoing activity in only a small subset. We then combined afferent calcium imaging with behavioral tracking of locomotion and estimates of local meningeal deformations. This approach revealed that CSD causes prolonged augmentation of afferent responsiveness to meningeal deformations associated with locomotion in previously sensitive afferents and emergent mechanical responses in previously silent afferents. These data support the notion that enhanced responsiveness of meningeal afferents to local meningeal deformation is the neural substrate for headache pain associated with physical activity following migraine onset.

The current study represents the first characterization of a CSD-associated calcium wave across meningeal afferent fibers and along the length of individual fibers. A rise in intracellular calcium detected by the GCaMP sensor normally indicates an action potential-evoked calcium influx (*Chen et al., 2013*). In contrast, the seconds-long wave of calcium elevation along individual afferent fibers we observed is incongruent with the generation of action potentials. It may instead be related to subthreshold depolarizations (*Li et al., 2022*) and the opening of voltage-gated calcium channels (*Awatramani et al., 2005*). Our data thus support the notion that meningeal afferents can generate spatially localized and subthreshold yet robust calcium transients during CSD. In turn, increased intracellular calcium could drive the release of sensory neuropeptides, such as CGRP, that can promote a local neurogenic inflammatory response linked to migraine pain (*Akerman et al., 2003*; *Amrutkar et al., 2011*; *Levy and Moskowitz, 2023*).

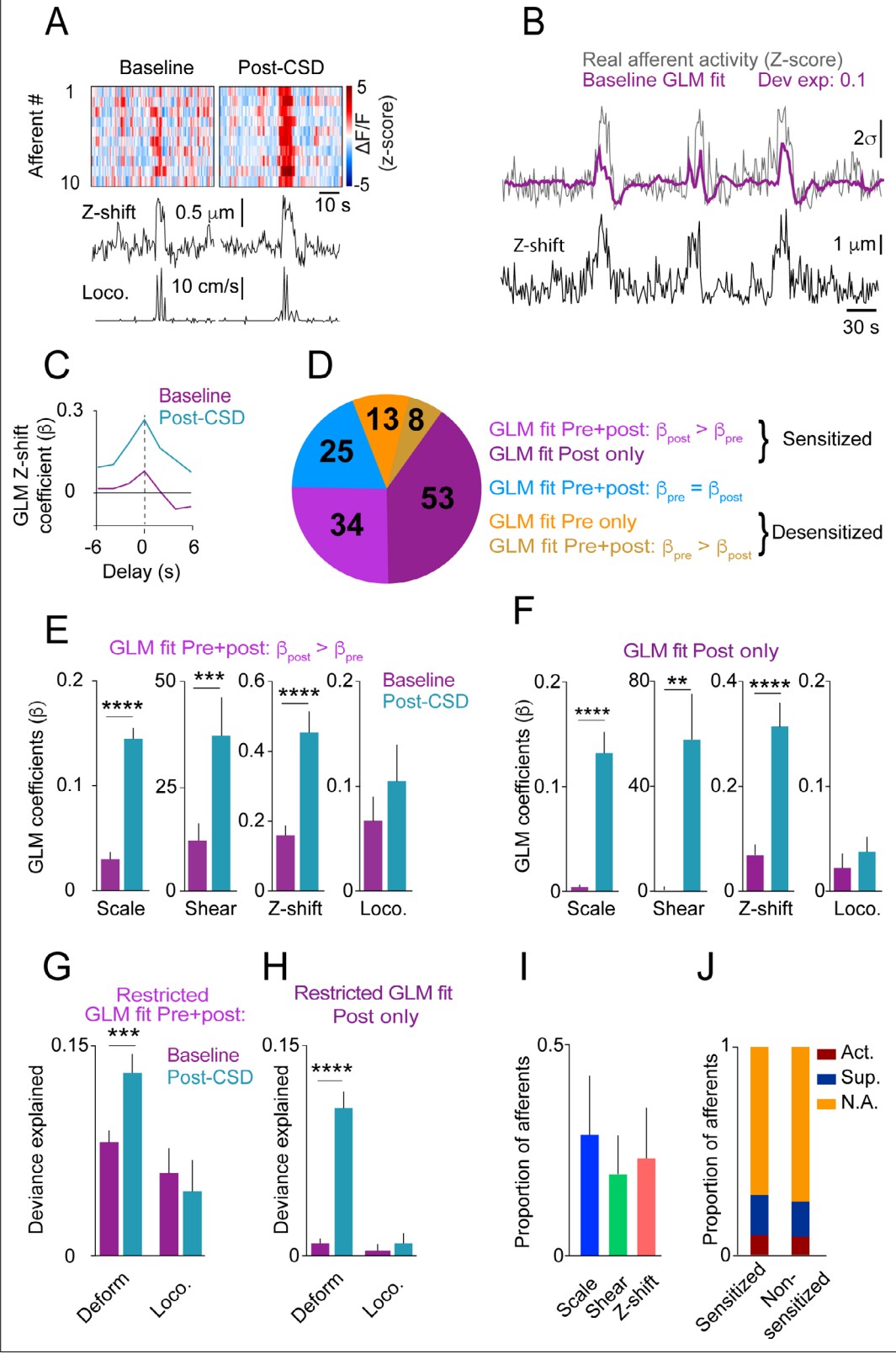

**Figure 4.** Cortical spreading depolarization (CSD) leads to sensitization of meningeal afferents to local deformation signals. (**A**) Example of meningeal afferent sensitization following CSD. Locomotion and its related Z-shift (bottom traces) are comparable before (left) and after (right) CSD, but afferent fibers exhibit greater responses associated with the Z-shift after CSD (heatmaps, top panels). (**B**) Example general linear model (GLM) fit

*Figure 4 continued on next page*

*Figure 4 continued*

of afferent activity in response to Z-shifts before CSD. A raw calcium activity trace recorded pre-CSD (gray traces, Z-scored; σ: 1 standard deviation) is plotted along with the model fit (purple). The deviance explained ('dev exp') is a metric of GLM fit quality and is above the threshold (0.05) for classifying an afferent's activity as reasonably well fit by the GLM. The activity of this example afferent could not be predicted by other deformation or locomotion variables (not shown), suggesting unique sensitivity to Z-shift. (**C**) GLM β coefficients used as a metric of the coupling between the Z-shift and the activity of the example afferent shown in b across different delays. A maximal coefficient at zero delay indicates the alignment of activity with Z-shifts. Note the greater afferent activation per unit Z-shift after CSD relative to baseline, indicative of an augmented or sensitized response. (**D**) Pie chart indicating the numbers and distribution of all afferents well fit by deformation and/or locomotion signals either before and/or after CSD. Afferents were categorized as sensitized if they (i) had significant GLM fits both pre- and post-CSD and higher coefficients for a given deformation and/or locomotion predictor post-CSD (purple) or (ii) were well fit only post-CSD (magenta). Two small subsets of afferents categorized as desensitized had worse GLM fits post-CSD (mustard) or were no longer well fit post-CSD (orange). The incidence of afferent sensitization exceeded that of desensitization (p<0.001, $X^2$ test). (**E, F**) Comparisons of pre- and post-CSD GLM coefficients for the deformation and locomotion predictors. Data are shown for sensitized afferents with well-fit models pre- and post-CSD (corresponding to the purple population in d) and for afferents with well-fit models only post-CSD (i.e. silent pre-CSD, corresponding to the magenta population in d). Mouse velocity coefficients were close to 0 in all cases (not shown). In the two sensitized afferent populations, only coefficients related to deformation predictors increased post-CSD (**p<0.01, ***p<0.001, ****p<0.0001, Wilcoxon sign rank test with correction for multiple analyses). (**G, H**) The response bias of sensitized afferents to meningeal deformation was further observed when comparing these GLMs to restricted GLMs that included only the group of deformation predictors or the group of locomotion predictors. The deviance explained by the deformation response component (estimated as the difference between the full GLM and the GLM lacking deformation variables) was significantly greater than for the locomotion response component in sensitized afferents that were well fit pre- and post-CSD and for those that were well fit only post-CSD (***p<0.001 and ****p<0.0001, Wilcoxon test for g and h, respectively). Bars depict mean; error bars indicate SEM. (**I**) Among the sensitized afferents with enhanced sensitivity to deformation variables, we observed a similar sensitization to scale, shear, and Z-shift variables. Bars depict mean; error bars indicate SEM. (**J**) There was no difference in the incidence of sensitized afferents among afferents that showed prolonged activation, prolonged suppression, or no change in ongoing activity post-CSD (p=0.9, $X^2$ test; *Figure 2*).

The mechanisms underlying the acute CSD-related afferent calcium wave remain unclear. We observed short-lasting meningeal deformation in response to CSD, likely due to the swelling of cortical cells and decreased extracellular space. However, this deformation was not associated with the acute afferent response, thus unlikely to be its primary driver. A mechanism involving local depolarizing effects of diffusible excitatory molecules, such as potassium ions, whose cortical levels show a wave of elevation coincident with the propagation of the CSD wave (*Suryavanshi et al., 2022*) is more likely to play a role. Since we also observed instantaneous elevation in calcium activity across subregions of individual afferent fibers oriented perpendicular to the calcium wave, we cannot exclude the possibility that some afferents also signal via action potentials during the passage of the CSD wave.

Electrophysiological recordings in anesthetized rats previously demonstrated prolonged elevations in spiking in ~50% of the trigeminal somata of meningeal afferents following CSD. In most recordings, increased activity emerged after an ~10 min delay and persisted for nearly an hour following CSD (*Zhao and Levy, 2015*). In the awake mouse, the propensity for these prolonged afferent responses was considerably smaller and their duration was notably shorter, raising questions about the relevance of this response to migraine pain. While species differences and the effects of anesthesia in the rat studies might play a role, meningeal irritation due to the acute large craniotomies used in previous electrophysiological recording studies could be a major contributing factor. Craniotomy increases meningeal permeability (*Roth et al., 2014*; *Zhao et al., 2017*), which could facilitate the transfer of algesic signals from the cortex. An acute craniotomy also leads to a meningeal inflammatory response via the activation of local immune cells (*Levy et al., 2007*) and could prime meningeal afferents to develop prolonged activation following CSD. If such a priming mechanism occurs in susceptible individuals who suffer from migraines, it could facilitate the activation of meningeal afferents and drive the headache during a migraine attack. These processes are likely less prevalent when using our chronic cranial window approach, as we have previously shown that this approach is not associated with meningeal vascular and cortical inflammation (*Goldey et al., 2014*; *Blaeser et al., 2022d*). The

very low level of ongoing activity (fluorescent events) we observed at baseline also suggests a lack of an inflammatory response and associated afferent priming before CSD.

Unexpectedly, a separate afferent population exhibited suppressed calcium activity that began immediately after the acute response and lasted approximately 25 min. This pattern of afferent responses resembles the rapid loss of cortical activity in the wake of CSD. However, its faster recovery points to a mechanism distinct from the pre- and post-synaptic changes responsible for the silencing of cortical activity (*Sawant-Pokam et al., 2017*; *Kucharz and Lauritzen, 2018*). Whether CSD drives the local release of mediators capable of reducing the excitability and spiking dynamics in a subset of afferents will require further studies.

A major finding of this study is the post-CSD development of augmented meningeal afferent responsiveness to meningeal deformations associated with locomotion bouts in awake mice with closed and pressurized meninges. We previously found that the activity of most meningeal afferents around the time of locomotion is driven by mixed sensitivities to meningeal deformation signals and to the binary state of locomotion. However, the activity of about a third of the afferents is more closely associated with the moment-to-moment deformation than with the state of locomotion per se (*Blaeser et al., 2022d*). Here, we demonstrate that the amplification of afferent responsiveness following CSD relates primarily to enhanced neural gain in response to meningeal deformation rather than to other physiological processes associated with locomotion, such as cerebral vasodilation and a minor increase in intracranial pressure (*Gao and Drew, 2016*). This increased sensitivity suggests that the afferent population sensitive to meningeal deformation has unique properties that render it most susceptible to becoming sensitized following CSD. Migraine headaches are often worsened by abrupt maneuvers, such as coughing and sneezing, resulting in a larger increase in intracranial pressure than during locomotion. Thus, our observation of heightened responsiveness to locomotion-related meningeal deformation may underestimate the increased afferent responsivity post-CSD during other behaviors, such as coughing.

Meningeal afferents responding to physiological meningeal deformations may constitute a population of low-threshold mechanoreceptor (LTMR) afferents *von Buchholtz et al., 2020* whose activation under normal conditions is unlikely to produce headache. Our finding that CSD can augment the mechanical sensitivity of these afferents suggests that they may also possess nociceptive properties. The discovery of a subset of cutaneous A-LTMR trigeminal afferents that also respond to noxious mechanical stimuli (*von Buchholtz et al., 2021*) supports this view. Our data further suggest that about half of all afferents deemed to be sensitized following CSD are likely higher-threshold mechanosensitive afferents, as they were not driven by meningeal deformations or locomotion at baseline. Accordingly, local inflammation, which occurs following CSD, could recruit 'silent' meningeal nociceptive afferents to become functional mechanonociceptors (*Strassman and Levy, 2006*). Overall, we propose that the sensitization of silent nociceptors, as well as of afferents with basal responsiveness to acute meningeal deformations, could produce a state of intracranial mechanical allodynia that underlies the exacerbation of migraine headaches during physical exertion and associated meningeal deformations.

## Materials and methods

### Animals

All experimental procedures complied with the ARRIVE and were approved by the Beth Israel Deaconess Medical Center Institutional Animal Care and Use Committee (protocols #105-2015, 072-2021). All experiments were conducted on adult (8–16 weeks of age) C57BL/6J mice (5 males, 2 females, Jackson Laboratory). Mice were group-housed with standard mouse chow and water provided ad libitum before viral injection (see below). Mice used for in vivo two-photon imaging were singly housed and provided a running wheel, a hut, and a chew bar.

### Surgical procedures and CSD induction

All surgical procedures were performed in anesthetized mice (isoflurane in $O_2$; 3.5% for induction, 1.5% for maintenance). Animals were given Meloxicam SR (4 mg/kg s.c.) for post-surgical analgesia. For monitoring calcium activity in meningeal afferents, 2.0 µl of AAV2/5.CAG.GCaMP6s.WPRE.SV40 (titer: $1 \times 10^{13}$; Addgene, 100844-AAV5; RRID:100844) was injected into the left trigeminal ganglion

(TG) using the following stereotaxic coordinates: 1.5 mm lateral and 0.3–0.8 mm anterior to Bregma and 7.0–7.2 mm ventral to the dura at a lateral-to-medial tilt with an angle of 22.5° relative to the dorsal-ventral axis. We previously verified this approach by examining GCaMP6s expression in TG somata and meningeal afferent fibers 8 weeks after injection using immunohistochemistry (*Blaeser et al., 2022d*). Mice used for in vivo two-photon imaging were instrumented with a titanium headpost and a 3 mm cranial window (*Goldey et al., 2014*; *Blaeser et al., 2022d*) covering the posterior cortex (window centered roughly 1.5 mm lateral and 2 mm posterior to Bregma over the left hemisphere) 6–8 weeks after AAV injection. For CSD induction, a burr hole (~0.5 mm diameter) was drilled 1.5 mm anterior to the edge of the window until the brain's surface was barely visible. The burr hole was then plugged using a silicone elastomer (Kwik-Cast, WPI), and the mouse was allowed to recover. To trigger a single CSD episode, a glass micropipette (50 µm diameter) was briefly inserted through the silicon plug ~1 mm deep into the cortex for 2 s (*Zhao and Levy, 2016*).

## Running wheel habituation

After at least 1 week of recovery following cranial window implantation, mice were head-fixed on a 3D-printed running wheel for gradual habituation (10 min to 1 hr over 3–4 days). To minimize downward forces on the bone and meninges produced while the mouse pushed upward against the headpost, the running wheel was mounted on a cantilever (*Ramesh et al., 2018*; *Blaeser et al., 2022d*). The position of the headpost, anterior to where the mouse's paws touch the wheel, also made it hard for the mouse to push straight up and apply forces to the skull. The strong cement used to bind all skull plates and headpost together (*Goldey et al., 2014*) further mitigated any movement-induced strain on the skull that might affect the underlying meninges. Mice displaying signs of stress were immediately removed from head fixation, and additional habituation days were added until mice tolerated head fixation without visible signs of stress. Mice received a high-calorie liquid meal replacement (Ensure) via a syringe as part of the habituation process.

## Two-photon imaging

Calcium imaging was performed as recently described (*Blaeser et al., 2022d*) while mice were head-fixed on the running wheel. We used a Nikon 16X, 0.8 NA water immersion objective on a resonant-scanning two-photon microscope (Neurolabware) and a MaiTai DeepSee laser, set to 920 nm with 25–40 mW power for GCaMP6s visualization. Digital zoom was set at 2.4× (626×423 µm$^2$ FOVs). In seven experiments, we imaged across a 60 µm volume (3D) using an electrically tunable lens (Optotune) at 1.03/s. We employed volumetric imaging for three main reasons: (1) To capture the activity of afferents throughout the meningeal volume. In our volumetric imaging approach, including in this work, we observed afferent calcium signals throughout the meningeal thickness (see Figure 5 in *Blaeser et al., 2022d*). However, the majority of afferents were localized to the most superficial 20 µm (Figure S1E in *Blaeser et al., 2022d*), suggesting that we mostly recorded the activity of dural afferents; (2) to enable simultaneous quantification of 3D deformations and the activity of afferents throughout the thickness of the meninges. This allowed us to determine whether changes in mechanosensitivity could involve augmented activity in response to intracranial mechanical forces that produced meningeal deformation along the Z-axis of the meninges (e.g. increased intracranial pressure); (3) to provide a direct means to confirm that the afferent GCaMP fluorescent changes we observed were not due to artifacts related to meningeal motion along the Z-axis. In two experiments, only single-plane (2D) data were collected at the cranial dura level at 15.5 Hz due to technical issues. In every experiment, we conducted two imaging runs (30 min each) to collect baseline data, followed by four more 30 min runs post-CSD induction. In a subset of experiments in which the FOV included a visible large pial artery (n=4), we verified the induction of a CSD by visualizing its vascular signature, including a brief vasoconstriction followed by dilation (*Rosic et al., 2019*; *Figure 1—figure supplement 1A*).

## Locomotion signals

Wheel position during each imaging run was recorded using an Arduino Uno board at 15.5 Hz. The instantaneous velocity was calculated as the time derivative of this signal and was downsampled to match the sampling rate of volume scans. Locomotion state was determined using a two-state hidden Markov model. Locomotion bouts were defined as periods when the locomotion state was sustained for at least 2 s (*Blaeser et al., 2022d*).

## Image processing and calcium signal extraction

All image processing and analyses were performed in MATLAB 2020a and ImageJ (Fiji, NIH) as described previously (*Shipley et al., 2020*; *Blaeser et al., 2022d*). In brief, imaging movies were subjected to several preprocessing steps. For single-plane experiments, all imaging runs were concatenated into a single movie. A reference image was defined as the mean projection over the middle 50% of frames from the second run pre-CSD. Each frame of the concatenated movie was then affine registered to the reference using the TurboReg plugin for ImageJ. For volumetric imaging, we first corrected the lensing effect due to the electrically tunable lens. We next subjected the movies to rigid registration, using the discrete Fourier transform to correct for within-volume (2D) translations and translations along the z-axis. We next used z-interpolation to correct for z-translations of individual imaged planes within each volume. The last step involved affine registration using a reference volume formed from the mean projection of the middle 50% of frames from the second run.

Registered 2D movies were analyzed using a PCA/ICA package (*Mukamel et al., 2009*) to extract masks of pixels with correlated activity. Users screened each prospective ROI for morphology and fluorescence signal quality. Afferent ROIs with <50 pixels were rejected. For each ROI included in subsequent analyses, we generated a dilated mask extending 8–21 pixels from the outer edge of the ROI, excluding any pixels that belonged to another ROI. This 'neuropil' mask was used for the subtraction of background signals. In volumetric imaging, we generated a mean projection over all planes containing afferents for each volume. Then, we ran the resulting 2D movie through the PCA/ICA procedure, yielding an initial set of 2D masks representing putative 3D ROIs. An initial fluorescence trace extracted from each 2D mask was calculated by averaging fluorescence across all pixels in the mask. To identify which voxels in the original volumetric dataset contributed most strongly to each fluorescence trace, we calculated the Pearson correlation of this trace with the fluorescence time course of each voxel, resulting in a 3D volume of correlation values. These 3D correlation volumes were then screened manually for quality of morphology and signal. The surviving volumes were thresholded at the 75th percentile of correlation across all voxels to form putative 3D ROI masks.

## Fluorescence signals

We calculated raw fluorescence signals at each time point for the ith ROI ($F^i_{ROI}$) and its corresponding neuropil mask ($F^i_{np}$), as the simple arithmetic means of all pixels/voxels within each ROI mask. Next, we calculated $F^i = F^i_{ROI} - F^i_{np} + <F^i_{np}>$, where brackets denote the mean across the entire recording. We then calculated the corresponding baseline signal $F^i_0$ as the 10th percentile of a moving window for the last 32 s (*Sugden et al., 2020*). We then calculated the normalized, baseline-subtracted time series $\Delta F/F_0 = (F^i - F^i_0)/F^i_0$. This signal was standardized akin to a Z-score operation by subtracting the median value and dividing by the standard deviation (calculated during quiet wakefulness, an epoch with low levels of evoked activity). Fluorescence events were defined as periods where the signal consistently exceeded a value of 1 for at least 1 s and where peak fractional change in fluorescence ($\Delta F/F_0$) was at least 5%.

## Identifying calcium activity in afferent fibers

To analyze calcium activity related to an afferent fiber, sets of ROIs putatively belonging to the same axon were initially identified using a previously described method (*Liang et al., 2018*; *Blaeser et al., 2022d*). We calculated the pairwise fluorescence event correlation between ROIs during quiet wakefulness, thresholding at 0.7 correlation. We then calculated the cosine dissimilarity between the full set of correlation coefficients for each pair of ROIs, which was used to calculate the linkage between each pair. Finally, hierarchical clustering was performed using a cutoff value of 2. This procedure generated sets of ROIs that were mutually highly correlated. We then visually inspected each cluster and manually identified the subsets of ROIs that unambiguously covered the same specific afferent fiber without any branching. We used the mean activity of these ROI subsets to analyze each afferent's calcium activity.

## CSD-associated calcium wave characterization

To detect the CSD-associated meningeal calcium wave (see *Figure 1—figure supplement 1B*), we first calculated the mean fluorescence signal over all voxels in the FOV, $F_{FOV}(t)$, and its time derivative $dF_{FOV}/dt$, in the first 5 min after cortical pinprick. To identify the timing of the CSD, we first defined an

initial period, $t_{inital}$, between the derivative's maximum and minimum. We then calculated baseline fluorescence, $F_{pre}$, as the 10th percentile value of $F_{FOV}$ in the 30 s before the initial period and a normalized fluorescence signal, $\Delta F/F_{pre} = (F_{FOV} – F_{pre})/F_{pre}$. Next, we defined a threshold value as $0.1*max(\Delta F(t_{initial})/F_{pre})$. The wave's final onset and offset times were defined as the points around the peak where $\Delta F_{FOV}/F_{pre}$ crossed the threshold value.

To measure the propagation of the calcium wave throughout the meninges, we estimated the wave's onset time at different X and Y positions within the FOV. Briefly, we gathered all the data from the peri-CSD wave from –6 to 14 s relative to the calcium wave onset time determined above. All voxels belonging to or neighboring (within 8 pixels) ROIs were excluded to focus on the overall wave-like advancement in the background 'neuropil' fluorescence signal (which integrates background fluorescence across depths and thus reflects a smooth estimate of mean activity in a given region). The resulting series of images was then divided into 40×40 pixel spatial bins, and mean fluorescence signals $F_{np}^{bin}(t)$ were calculated. The time of estimated onset of the CSD-associated wave at a given bin was estimated by fitting $F_{np}^{bin}(t)$ to a logistic function, $A/(1+exp(-(t-t_{onset})/\tau))+K$. Fits with $R^2<0.5$, or $\tau >2$ s, or $\tau <0$ s, were excluded. The wave's speed was estimated by linear regression of onset times as a function of distance. The direction of the wave was determined by estimating the contour of the wavefront at the median bin onset time, fitting this contour to a line, and then calculating the vector orthogonal to that line.

To specifically examine the propagation of the fluorescence signals along long (>200 μm) individual afferent fibers, we determined the timing of CSD-associated wave onset by fitting each ROI's peri-wave fluorescence signals to a sigmoidal function and then estimating the speed of propagation by linear regression of onset times as a function of distance. For comparison, this procedure was repeated using the peri-locomotion bout signals. Since locomotion-associated activation occurred essentially simultaneously (i.e. $\Delta t=0$ s), even for distant ROIs, we report these results in terms of the pace (the inverse of speed) to avoid dividing by zero.

## Assessment of CSD-evoked changes in afferent ongoing activity

To determine changes in ongoing afferent activity, we focused on the periods of immobility between locomotion bouts (stillness). We minimized any residual effects of locomotion-related activity by excluding 30 s epochs before and following the locomotion bouts. We then analyzed fluorescence events as above. We estimated levels of ongoing activity rate from the normalized Z-score time courses as the fraction of time the afferents exhibited fluorescent events (defined as above) during each 1 min interval. We defined afferents with post-CSD increases or suppression of activity if changes in ongoing activity lasted >10 consecutive min and began within 30 min following the CSD wave (*Zhao et al., 2021*).

## Meningeal deformation signals

To estimate the degree of meningeal deformation in locomoting mice, we extracted the values produced from the affine registration procedure used to correct intra-volume (XY) and inter-volume (Z) image displacements (see above). Corrections made to account for XY displacements were used to quantify the amount of scaling (expansion and compression) and shearing. Corrections made along the z-axis measured how much each plane moved up or down relative to a reference volume ('Z-shift'). Scaling and Z-shifts were converted from pixels to microns. Positive Z-shifts indicate meningeal movement toward the skull.

## General linear models

To classify afferent responses, we fit Gaussian GLMs for locomotion, deformation, and fluorescence using the glmnet package in MATLAB. To allow for the possibility of a delay between the predictor and response, we expanded the set of predictors to include variables at varying delays relative to the calcium activity (*Driscoll et al., 2017*; *Ramesh et al., 2018*; *Blaeser et al., 2022d*). Specifically, we generated a set of temporally shifted versions for each variable spanning a time window from –6 to +6 s. These sets of arrays of temporal shifts for each variable were joined to form an array of temporally shifted predictor signals. The GLM was fit on 75% of the data for each cell with elastic net regularization (α=0.01). We then used the GLM coefficients to measure the deviance explained on the remaining 25% of the data. The relative explanatory value of locomotion or deformation variables

was calculated by refitting the GLM after excluding the family of predictors (locomotion state and velocity for evaluating the effect of locomotion and scale, shear, and Z-shift for examining the effect of deformation) and calculating the difference in deviance explained by the full model versus the model lacking a given family of predictor variables. All GLMs underwent 10-fold cross-validation.

## Data analysis and statistics

Data analyses were performed in MATLAB 2020a and Prism 9. Sample sizes were not predetermined by power analysis but are similar to previous studies (*Sugden et al., 2020*; *Blaeser et al., 2022d*). Two-tailed paired t-tests and one-way analysis of variance followed by a post hoc Tukey's test were used for all parametric data. Data with non-Gaussian distributions were analyzed using a Wilcoxon matched-pairs signed rank sum test or a Mann-Whitney U-test. Corrections for multiple comparisons were adjusted using the false discovery rate approach. Bootstrapped confidence intervals and hypothesis tests were generated using the 'iboot' iterated bootstrapping package (*Penn, 2020*). Unless otherwise noted, data are presented as averages ± standard error of the mean (SEM). P-Values are indicated as follows: $p < 0.05$ (*), $p < 0.01$ (**), $p < 0.001$ (***), $p < 0.0001$ (****).

## Acknowledgements

We thank members of the Andermann and Levy labs for helpful discussions, and Fred Shipley, Glenn Goldey, Kiersten M Levandowski, Helaine Gariepy, Andrew Lutas, and Osama Alturkistani for advice and technical assistance. We thank Drs. Jayaraman, Kerr, Kim, Looger, and Svoboda and the GENIE Project, Janelia Farm Research Campus, HHMI, for GCaMP6s. Support was provided by NIH DP2DK105570, R01DK109930, DP1AT010971, the Pew Innovation Fund (to MLA); R21NS101405 (to DL and MLA); R01NS086830; R01NS078263; R01NS115972 (to DL); and T32 5T32DK007516 (AUS).

## Additional information

### Funding

| Funder | Grant reference number | Author |
|---|---|---|
| National Institutes of Health | DP2DK105570 | Mark L Andermann |
| National Institute of Diabetes and Digestive and Kidney Diseases | R01DK109930 | Mark L Andermann |
| National Institutes of Health | DP1AT010971 | Mark L Andermann |
| Pew Charitable Trusts | Pew Innovation Fund | Mark L Andermann |
| National Institute of Neurological Disorders and Stroke | R21NS101405 | Dan Levy Mark L Andermann |
| National Institute of Neurological Disorders and Stroke | R01NS086830 | Dan Levy |
| National Institute of Neurological Disorders and Stroke | R01NS078263 | Dan Levy |
| National Institute of Neurological Disorders and Stroke | R01NS115972 | Dan Levy |
| National Institute of Diabetes and Digestive and Kidney Diseases | T32 5T32DK007516 | Arthur U Sugden |

The funders had no role in study design, data collection and interpretation, or the decision to submit the work for publication.

## Author contributions
Andrew S Blaeser, Data curation, Investigation, Methodology; Jun Zhao, Arthur U Sugden, Data curation, Software, Formal analysis, Investigation, Methodology; Simone Carneiro-Nascimento, Mark L Andermann, Conceptualization, Formal analysis, Supervision, Funding acquisition, Investigation, Methodology, Writing - original draft, Project administration, Writing - review and editing; Dan Levy, Conceptualization, Formal analysis, Supervision, Funding acquisition, Investigation, Writing - original draft, Project administration, Writing - review and editing

## Author ORCIDs
Mark L Andermann ⓘ http://orcid.org/0000-0002-9882-933X
Dan Levy ⓘ http://orcid.org/0000-0003-0630-6660

## Ethics
All experimental procedures complied with the ARRIVE and were approved by the Beth Israel Deaconess Medical Center Institutional Animal Care and Use Committee (protocols #105-2015, 072-2021).

Reviewer #1 (Public Review): https://doi.org/10.7554/eLife.91871.3.sa1
Reviewer #2 (Public Review): https://doi.org/10.7554/eLife.91871.3.sa2
Reviewer #3 (Public Review): https://doi.org/10.7554/eLife.91871.3.sa3
Author Response https://doi.org/10.7554/eLife.91871.3.sa4

# Additional files

## Supplementary files
• MDAR checklist

## Data availability
The code used to analyze locomotion data is available at GitHub (copy archived at *Blaeser, 2022a*). Code for organizing and processing two-photon imaging data is available at GitHub (copy archived at *Blaeser, 2022b*). The code for analysis of calcium imaging is available at GitHub (copy archived at *Blaeser, 2022c*). Any additional information required to reanalyze the data reported in this paper is available from the lead contact upon request.

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
