## [Editor Report · eLife assessment]

This **fundamental** study explored the impact of migraine-related cortical spreading depression (CSD) on the firing of nerves innervating the coverings of the brain that are considered the putative source of migraine-related pain. Using **convincing** approaches they show that these responses are altered in response to mechanical deformation of the brain coverings. Given that migraine is characterized by worsening head pain in response to movement, the findings offer a potential mechanism that may explain this clinical phenomenon.

---

## [Referee Report · Reviewer #1 (Public Review)]

Summary:

Herein, Blaeser et al. explored the impact of migraine-related cortical spreading depression (CSD) on the calcium dynamics of meningeal afferents that are considered the putative source of migraine-related pain. Critically previous studies have identified widespread activation of these meningeal afferents following CSD; however, most studies of this kind have been performed in anesthetized rodents. By conducting a series of technically challenging and compelling calcium imaging experiments in conscious head fixed mice they find in contrast that a much smaller proportion of meningeal afferents are persistently activated following CSD. Instead, they identify that post-CSD responses are differentially altered across a wide array of afferents, including increased and decreased responses to mechanical meningeal deformations and activation of previously non-responsive afferents following CSD. Given that migraine is characterized by worsening head pain in response to movement, the findings offer a potential mechanism that may explain this clinical phenomenon.

Strengths:

Using head fixed conscious mice overcomes the limitations of anesthetized preps and the potential impact of anaesthesia on meningeal afferent function which facilitated novel results when compared to previous anesthetized studies. Further, the authors used a closed cranial window preparation to maximize normal physiological states during recording, although the introduction of a needle prick to induce CSD will have generated a small opening in the cranial preparation, rendering it not fully closed as suggested. However, technical issues with available AAV's and alternate less invasive triggering methodologies necessitate the current approach.

Weaknesses:

Although this is a well conducted technically challenging study that has added valuable knowledge on the response of meningeal afferents the study would have benefited from the inclusion of more female mice. Migraine is a female dominant condition and an attempt to compare potential sex-differences in afferent responses would undoubtedly have improved the outcome. The authors report potential sex-specific effects on AAV transfection rates between males and females which have contributed to this imbalance.

The authors imply that the current method shows clear differences when compared to older anaesthetized studies; however, many of these were conducted in rats and relied on recording from the trigeminal ganglion. Attempts to address this point have proven difficult due to limited GCaMP signalling in anaesthetised mice, meaning that technical differences cannot be ruled out.

---

## [Referee Report · Reviewer #2 (Public Review)]

This is an interesting study examining the question of whether CSD sensitizes meningeal afferent sensory neurons leading to spontaneous activity or whether CSD sensitizes these neurons to mechanical stimulation related to locomotion. Using two-photon in vivo calcium imaging based on viral expression of GCaMP6 in the TG, awake mice on a running wheel were imaged following CSD induction by cortical pinprick. The CSD wave evoked a rise in intracellular calcium in many sensory neurons during the propagation of the wave but several patterns of afferent activity developed after the CSD. The minority of recorded neurons (10%) showed spontaneous activity while slightly larger numbers (20%) showed depression of activity, the latter pattern developed earlier than the former. The vast majority of neurons (70%) were unaffected by the CSD. CSD decreased the time spent running and the numbers of bouts per minute but each bout was unaffected by CSD. There also was no influence of CSD on the parameters referred to as meningeal deformation including scale, shear, and Z-shift. Using GLM, the authors then determine that there there is an increase in locomotion/deformation-related afferent activity in 51% of neurons, a decrease in 12% of neurons, and no change in 37%. GLM coefficients were increased for deformation related activity but not locomotion related activity after CSD. There also were an increase in afferents responsive to locomotion/deformation following CSD that were previously silent. This study shows that unlike prior reports, CSD does not lead to spontaneous activity in the majority of sensory neurons but that it increases sensitivity to mechanical deformation of the meninges. This has important implications for headache disorders like migraine where CSD is thought to contribute to the pathology in unclear ways with this new study suggesting that it may lead to increased mechanical sensitivity characteristic of migraine attacks.

---

## [Referee Report · Reviewer #3 (Public Review)]

Summary: In this manuscript, Blaeser et al. explore the link between CSD and headache pain. How does an electrochemical wave in the brain parenchyma, which lacks nociceptors, result in pain and allodynia in the V1-3 distribution? Prior work had established that CSD increased the firing rate of trigeminal neurons, measured electrophysiologically at the level of the peripheral ganglion. Here, Blaeser et al. focus on the fine afferent processes of the trigeminal neurons, resolving Ca2+ activity of individual fibers within the meninges. To accomplish these experiments, the authors injected AAV encoding the Ca2+ sensitive fluorophore GCamp6s into the trigeminal ganglion, and 8 weeks later imaged fluorescence signals from the afferent terminals within the meninges through a closed cranial window. They captured activity patterns at rest, with locomotion, and in response to CSD. They found that mechanical forces due to meningeal deformations during locomotion (shearing, scaling, and Z-shifts) drove non-spreading Ca2+ signals throughout the imaging field, whereas CSD caused propagating Ca2+ signals in the trigeminal afferent fibers, moving at the expected speed of CSD (3.8 mm/min). Following CSD, there were variable changes in basal GCamp6s signals: these signals were unchanged in the majority of fibers, signals increased (after a ~20 min delay) in 10% of fibers, and signals decreased in 20% of fibers. Bouts of locomotion were less frequent following CSD, but when they did occur, they elicited more robust GCamp6s signals than pre-CSD. These findings advance the field, suggesting that headache pain following CSD can be explained on the basis of peripheral cranial nerve activity, without invoking central sensitization at the brain stem/thalamic level. This insight could open new pathways for targeting the parenchymal-meningeal interface to develop novel abortive or preventive migraine treatments.

Strengths: The manuscript is well-written. The studies are broadly relevant to neuroscientists and physiologists, as well as neurologists, pain clinicians, and patients with migraine with aura and acephalgic migraine. The studies are well-conceived and appear to be technically well-executed.

Weaknesses: In the present study, conclusions are based entirely on fluorescence signals from GCamp6s. Fluorescence experiments should be interpreted cautiously in the context of CSD. GCamp6 fluorophores are strongly pH dependent, with decreased signal at acidic pH values (at matched Ca2+ concentration). CSD induces an impressive acidosis transient in the brain parenchyma, so one wonders whether the suppression of activity reported in the wake of CSD (Figure 2) in fact reflects decreased sensitivity of the GCamp6 reporter, rather than decreased activity in the fibers. If intracellular pH in trigeminal afferent fibers acidifies in the wake of CSD, GCamp6s fluorescence may underestimate the actual neuronal activity.

---

## [Author Response]

The following is the authors’ response to the original reviews.

We thank the reviewers for their thorough assessment of our study, their overall enthusiasm, and the helpful suggestions for clarifying the methods and results, additional analyses, and discussion points. We have made earnest efforts to address the weaknesses raised in the public review and other recommendations made by the reviewers.

**Public Reviews:**

**Reviewer #1 (Public Review):**
Herein, Blaeser et al. explored the impact of migraine-related cortical spreading depression (CSD) on the calcium dynamics of meningeal afferents that are considered the putative source of migraine-related pain. Critically previous studies have identified widespread activation of these meningeal afferents following CSD; however, most studies of this kind have been performed in anesthetized rodents. By conducting a series of technically challenging calcium imaging experiments in conscious head fixed mice they find in contrast that a much smaller proportion of meningeal afferents are persistently activated following CSD. Instead, they identify that post-CSD responses are differentially altered across a wide array of afferents, including increased and decreased responses to mechanical meningeal deformations and activation of previously non-responsive afferents following CSD. Given that migraine is characterized by worsening head pain in response to movement, the findings offer a potential mechanism that may explain this clinical phenomenon.Strengths:Using head fixed conscious mice overcomes the limitations of anesthetized preps and the potential impact of anaesthesia on meningeal afferent function which facilitated novel results when compared to previous anesthetized studies. Further, the authors used a closed cranial window preparation to maximize normal physiological states during recording, although the introduction of a needle prick to induce CSD will have generated a small opening in the cranial preparation, rendering it not fully closed as suggested.Weaknesses:Although this is a well conducted technically challenging study that has added valuable knowledge on the response of meningeal afferents the study would have benefited from the inclusion of more female mice. Migraine is a female dominant condition and an attempt to compare potential sex-differences in afferent responses would undoubtedly have improved the outcome.

Our study included only two females, largely reflecting the much higher success rate of AAV-mediated meningeal afferent GCaMP expression in males than in females. The reason for the lower yield in female mice is unclear to us at present but may involve, at least partly, sex-specific differences in the mechanisms responsible for efficient transduction with this AAV vector observed in peripheral tissues (Davidoff et al. 2003). While our study did not address sex differences, a recent study (Melo-Carrillo et al. 2017) reported CSD equally activating and sensitizing second-order dorsal horn neurons that receive input from meningeal afferents in male and female rats.

The authors imply that the current method shows clear differences when compared to older anaesthetized studies; however, many of these were conducted in rats and relied on recording from the trigeminal ganglion. Inclusion of a subgroup of anesthetized mice in the current preparation may have helped to answer these outstanding questions, being is this species dependent or as a result of the different technical approaches.

We have tried to address the anesthesia issue by conducting imaging sessions in several isoflurane-anesthetized mice. However, during these experiments, we observed a substantial decrease in the GCaMP fluorescence signal with a much lower signal-to-noise ratio that made the analyses of the afferents’ calcium signal unreliable. Reduced GCaMP signal in meningeal axons during anesthesia may be related to the development of respiratory acidosis, since lower pH leads to decreased GCaMP signal, as also mentioned by Reviewer #3. Of note, urethane anesthesia, which was used in all previous rat experiments, also produces respiratory acidosis.

The authors discuss meningeal deformations as a result of locomotion; however, despite referring to their previous work (Blaeser et al., 2022), the exact method of how these deformations were measured could be clearer. It is challenging to imaging that simple locomotion would induce such deformations and the one reference in the introduction refers to straining, such as cough that may induce intracranial hypertension, which is likely a more powerful stimulus than locomotion.

As part of the revision, we now provide a better description of the methodology (“Image processing and calcium signal extraction” section) used to determine meningeal deformations, including scaling, shearing, and Z-shift. In our previous paper (Blaeser et al. 2023), we provided an extensive description of the types of meningeal deformations occurring in locomoting mice. It should also be noted that locomotion drives cerebral vasodilation and intracranial pressure increases (Gao and Drew, 2016), which likely mediate, at least in part, the movement of the meninges towards the skull (positive Z-shift) and potentially other meningeal deformation parameters. We also agree with the reviewer that sudden maneuvers such as coughing and sneezing that lead to a larger increase in intracranial pressure are likely to be even more powerful drivers of endogenous intracranial mechanical stimulation than locomotion. Thus, our finding of increased responsiveness to locomotion-related meningeal deformation post-CSD may underestimate the increased afferent responsivity post-CSD during other behaviors such as coughing. We added this point to the discussion.

More recently, several groups have used optogenetic triggering of CSD to avoid opening of the cranium for needle prick. Given the authors robustly highlight the benefit of the closed cranium approach, would such an approach not have been more appropriate.

We agree with the reviewer that optogenetic methods used for CSD induction in non-craniotomized animals will further ensure accurate pressurization and, thus, will be an even better approach that avoids the burr hole used for pinprick. It should be noted, however, that the burr hole used for the pinprick likely had a minimal effect on intracranial pressure, as we minimized depressurization by plugging the burr hole throughout the experiments with a silicone elastomer. We have added this information to the revised Methods section.

It is also worth noting that the optogenetic methodology used by others to provoke CSD was optimized only recently and relies on transgenic mice with a strong expression of YFP (Thy1.ChR2-YFP mice) within the superficial cortex that is not compatible with the afferent GCaMP imaging of meningeal afferents. Modifications using red-shifted opsins may allow the use of this strategy in the future.

It was not clear how deformations predictors increased independent of locomotion (Figure 4D) as locomotion is essentially causing the deformations as noted in the study. This point was not so clear to this reviewer.

As noted in our previous paper (Blaeser et al., 2023), deformation variables often exhibit different time courses than locomotion, even when a deformation is initially induced by the onset of locomotion. Most notably, the scaling-related deformation ramps up slowly and often persists for tens of seconds after the onset and termination of locomotion, which may be related to the recovery dynamics of the meningeal vascular response to locomotion. Overall, while locomotion serves as a predictor of meningeal deformation, we observed previously (Blaeser et al. 2023) many afferents whose responses were more closely associated with the moment-to-moment deformations than with the state of locomotion per se, suggesting that a unique set of stimuli is responsible for the activation of this deformation-sensitive afferent population. The increased sensitivity to deformation signals we observed following CSD suggests that the afferent population sensitive to deformation has unique properties that render it most susceptible to becoming sensitized following CSD. We now discuss this possibility.

**Reviewer #2 (Public Review):**
This is an interesting study examining the question of whether CSD sensitizes meningeal afferent sensory neurons leading to spontaneous activity or whether CSD sensitizes these neurons to mechanical stimulation related to locomotion. Using two-photon in vivo calcium imaging based on viral expression of GCaMP6 in the TG, awake mice on a running wheel were imaged following CSD induction by cortical pinprick. The CSD wave evoked a rise in intracellular calcium in many sensory neurons during the propagation of the wave but several patterns of afferent activity developed after the CSD. The minority of recorded neurons (10%) showed spontaneous activity while slightly larger numbers (20%) showed depression of activity, the latter pattern developed earlier than the former. The vast majority of neurons (70%) were unaffected by the CSD. CSD decreased the time spent running and the numbers of bouts per minute but each bout was unaffected by CSD. There also was no influence of CSD on the parameters referred to as meningeal deformation including scale, shear, and Z-shift. Using GLM, the authors then determine that there there is an increase in locomotion/deformation-related afferent activity in 51% of neurons, a decrease in 12% of neurons, and no change in 37%. GLM coefficients were increased for deformation related activity but not locomotion related activity after CSD. There also was an increase in afferents responsive to locomotion/deformation following CSD that were previously silent. This study shows that unlike prior reports, CSD does not lead to spontaneous activity in the majority of sensory neurons but that it increases sensitivity to mechanical deformation of the meninges. This has important implications for headache disorders like migraine where CSD is thought to contribute to the pathology in unclear ways with this new study suggesting that it may lead to increased mechanical sensitivity characteristic of migraine attacks.1. It would be helpful to know what is meant by "post-CSD" in many of the figures where a time course is not shown. The methods indicate that 4, 30 min runs were collected after CSD but this would span 2 hours and the data do not indicate whether there are differences across time following CSD nor whether data from all 4 runs are averaged.

While we monitored time course changes in ongoing activity (see Figure 2), it was challenging to evaluate post-CSD changes in locomotion-related deformation responses at a fine temporal scale, as running bouts resumed at different time points post-CSD and occurred intermittently throughout the post-CSD analysis period. Our experiments were also not sufficiently powered to break out analyses at multiple different epochs post-CSD, partly because there wasn’t much locomotion. To allow comparisons using a sufficient number of bouts, we conducted our GLM analyses using all data collected during running bouts in the 2-hour post-CSD period (termed “post-CSD) versus in the 1-hour pre-CSD period. We have now clarified this further in the main text and figure legends.

1. Why is only the Z-shift data shown in Figures 4A-C? Each of the deformation values seems to contribute to the activity of neurons after CSD but only the Z-shift values are shown.

In many afferents, only one deformation variable best predicted the activity at both the pre- and post-CSD epochs. However, at the population level, all deformation variables were equally predictive. In the examples provided, the afferent developed augmented sensitivity that could only be predicted by the Z-shift variable, and the other deformation variables were not included to keep the figure legible. This is now clarified in the figure legend.

1. How much does the animal moving its skull against the head mount contribute to deformations of the meninges if the skull is potentially flexing during these movements? Even if mice are not locomoting, they can still attempt to move their heads thus creating pressure changes on the skull and underlying meninges. The authors mention in the methods that the strong cement used to bind the skull plates and headpost together minimize this, but how do they know it is minimized?

We did not measure skull flexing during locomotion and its potential effect on meningeal deformation. However, we would like to point out several considerations. It is evident from numerous imaging studies across various brain regions in freely moving animals, utilizing brain motion registration, that brain motion of the same scale (a few microns), as that observed in our studies, also occurs in the absence of head fixation (e.g., Glas et al, 2019; Zong et al 2021). In our system, the head-fixed mouse is locomoting on a cantilevered (spring-like) running wheel (see also Ramesh et al., 2018), which dissipates most, albeit not all, upward and forward forces applied to the skull during locomotion. Furthermore, the position of the headpost, anterior to where the mouse's paws touch the wheel, makes it hard for the mouse to push straight up and apply forces to the skull. We have updated the text in the methods section (Running wheel habituation) to address this. In our previous work (See Figure 2B in Blaeser et al. 2023), we found a substantial subset of afferents showing an increase in calcium activity that began after each bout of locomotion had terminated, and that lasted for many seconds, suggesting that skull flexing during locomotion may not play a leading role. Finally, we proposed in that study that meningeal deformations play a major role in the afferent response, given our findings of (i) sigmoidal stimulus-response curves between afferent activity and meningeal deformation and (ii) of different afferents that track scaling deformations along different axes. It is unlikely that all of these are related to any residual forces generated from skull deformations.

1. What is the mechanism by which afferents initiate the calcium wave during the CSD itself? Is this mechanical pressure due to swelling of the cortex during the wave? If so, why does the CSD have no impact on the deformation parameters? It seems that this cortical swelling would have some influence on these values unless the measurements of these values are taken well after cortical swelling subsides. Related to point 1 above, it is not clear when these measurements are taken post-CSD.

We provide, for the first time, evidence that CSD evokes local calcium elevation in meningeal afferent fibers in a manner that is incongruent with action potential propagation, as the activity gradually advances along individual afferents across many seconds during the wave. As indicated in Figure 1H, we measured these changes during the first 2 minutes post-CSD. Based on the reviewer’s question, we have now addressed whether mechanical changes occurring in the cortex in the wake of CSD might be responsible for the acute afferent activation we observed. We now include new data (Results, “Acute afferent activation is not related to CSD-evoked meningeal deformation” and Figure S2) showing an acute phase of meningeal deformation (as expected given the changes in extracellular fluid volume) lasting 40-80 seconds following the induction of CSD. Our data suggests, however, that these meningeal deformations are unlikely to be the main driver of the acute afferent calcium response. We propose that, based on the speed of the afferent calcium wave propagation and the distinct dynamics of calcium activity as compared to the dynamics of the deformations, the acute afferent response is more likely to be mediated by the spread of algesic mediators (e.g., glutamate, K+ ATP) and their diffusion into the overlying meninges.

Because the peri-CSD meningeal deformations return to baseline soon after the cessation of the CSD wave, they are unlikely to affect our analyses of post-CSD changes in afferent sensitivity in the following 2 hours. This is also supported by our data (see Figure 3F-H) showing similar locomotion-related deformations pre- and post-CSD, which were measured after the deformations related to the CSD itself had subsided.

1. How does CSD cause suppression of afferent activity? This is not discussed. It is probably a good idea in this discussion to reinforce that suppression in this case is suppression of the calcium response and not necessarily suppression of all neuronal activity.

The mechanism underlying the suppression of afferent activity remains unclear. We now discuss the following points:

First, the pattern of afferent responses resembles the rapid loss of cortical activity in the wake of a CSD, but its faster recovery points to a mechanism distinct from the pre-and post-synaptic changes responsible for the silencing of cortical activity (Sawant-Pokam et al., 2017; Kucharz and Lauritzen, 2018). Whether CSD drives the local release of mediators capable of reducing afferent excitability and spiking dynamics will require further studies.

Second, the reviewer proposes that the suppressed calcium activity we observed in ~20% of the afferents immediately following CSD may reflect a decreased calcium response independent of afferent spiking activity. Such a process could theoretically involve factors influencing the GCaMP fluorescence (see also our response to Reviewer #3) and/or factors modifying the afferents’ spiking-to-calcium coupling. We note that if a CSD-related factor could modify the calcium response independent of afferent spiking, one would expect a more consistent effect across axons, reflected as a reduced signal in a larger proportion of the afferents, which we did not observe.

1. How do the authors interpret the influence of CSD on locomotor activity? There was a decrease in bouts but the bouts themselves showed similar patterns after CSD. Is CSD merely inhibiting the initiation of bouts? Is this consistent with what CSD is known to do to motor activity? And again related to point 1, how long after CSD were these measurements taken? Were there changes in locomotor activity during the actual CSD compared to post-CSD?

To the best of our knowledge, there is very little data on the effect of CSD on motor activity, making it challenging to engage in further speculation regarding the mechanisms underlying the preservation of running bouts patterns post-CSD. Houben et al. (2017) described a similar reduction in locomotion in mice, corresponding to decreased motor cortex (M1) activity, and preservation of intermittent locomotion bouts. In the revised Results section, we now provide information about the cessation of locomotor activity during the CSD wave and have added information regarding the measurement of locomotion following CSD.

1. The authors mention the caveats of prior work where the skull is open and is thus depressurized. Is this not also the case here given there is a hole in the skull needed to induce CSD?

Unlike previous electrophysiological studies, which involved several large openings (~2x2 mm), including at the site of the afferents’ receptive field, our study involved only a small burr hole located remotely (1.5 mm) from the frontal edge of our imaging window. As noted in our response to Reviewer #1, this burr hole (~0.5 mm diameter) was unlikely to produce inflammation at the imaging site or cause depressurization as it was sealed with a silicone plug throughout the experiment.

1. The authors should check the %'s and the numbers in the pie chart for Figure 4. Line 224 says 53 is 22% but it does not look this way from the chart.

The 22% reported is the percentage of afferents that developed sensitivity post-CSD among all the non-sensitive ones pre-CSD. The pie chart illustrates only afferents that were deemed sensitive before and/or after the CSD. We removed the % to clarify.

1. Line 319 mentions that CSD causes "powerful calcium transients" in sensory neurons but it is not clear what is meant by powerful if there are no downstream effects of these transients being measured. The speculation is that these calcium transients could cause transmitter release, which would be an important observation in the absence of AP firing, but there are no data evaluating whether this is the case.

We changed the term to “robust”

**Reviewer #3 (Public Review):**
Summary:Blaeser et al. set out to explore the link between CSD and headache pain. How does an electrochemical wave in the brain parenchyma, which lacks nociceptors, result in pain and allodynia in the V1-3 distribution? Prior work had established that CSD increased the firing rate of trigeminal neurons, measured electrophysiologically at the level of the peripheral ganglion. Here, Blaeser et al. focus on the fine afferent processes of the trigeminal neurons, resolving Ca2+ activity of individual fibers within the meninges. To accomplish these experiments, the authors injected AAV encoding the Ca2+ sensitive fluorophore GCamp6s into the trigeminal ganglion, and 8 weeks later imaged fluorescence signals from the afferent terminals within the meninges through a closed cranial window. They captured activity patterns at rest, with locomotion, and in response to CSD. They found that mechanical forces due to meningeal deformations during locomotion (shearing, scaling, and Z-shifts) drove non-spreading Ca2+ signals throughout the imaging field, whereas CSD caused propagating Ca2+ signals in the trigeminal afferent fibers, moving at the expected speed of CSD (3.8 mm/min). Following CSD, there were variable changes in basal GCamp6s signals: these signals decreased in the majority of fibers, signals increased (after a 25 min delay) in other fibers, and signals remained unchanged in the remainder of fibers. Bouts of locomotion were less frequent following CSD, but when they did occur, they elicited more robust GCamp6s signals than pre-CSD. These findings advance the field, suggesting that headache pain following CSD can be explained on the basis of peripheral cranial nerve activity, without invoking central sensitization at the brain stem/thalamic level. This insight could open new pathways for targeting the parenchymal-meningeal interface to develop novel abortive or preventive migraine treatments.Strengths:The manuscript is well-written. The studies are broadly relevant to neuroscientists and physiologists, as well as neurologists, pain clinicians, and patients with migraine with aura and acephalgic migraine. The studies are well-conceived and appear to be technically well-executed.Weaknesses:1. Lack of anatomic confirmation that the dura were intact in these studies: it is notoriously challenging to create a cranial window in mouse skull without disrupting or even removing the dura. It was unclear which meningeal layers were captured in the imaging plane. Did the visualized trigeminal afferents terminate in the dura, subarachnoid space, or pia (as suggested by Supplemental Fig 1, capturing a pial artery in the imaging plane)? Were z-stacks obtained, to maintain the imaging plane, or to follow visualized afferents when they migrated out of the imaging plane during meningeal deformations?

We agree that avoiding disruption of the dura is challenging. Indeed, it took many months of practice before conducting the experiments in this manuscript to master methods for a craniotomy that spared the dura.

We addressed the issue of meningeal irritation due to cranial window surgery in our previous work (Blaeser et al., 2023). In brief, we conducted vascular imaging using the same cranial window approach and showed no leakage of macromolecules from dural or pial vessels anywhere within the imaging window at 2-6 weeks after the surgery (Figure S1D in Blaeser et al. 2022). This data suggested no ongoing meningeal inflammation below the window. The very low level of ongoing activity we observed at baseline also suggests a lack of an inflammatory response that could lead to afferent sensitization before CSD. This is now mentioned in the Discussion.

We conducted volumetric imaging for three main reasons: (1) To capture the activity of afferents throughout the meningeal volume. In our volumetric imaging approach, including in this work, we observed afferent calcium signals throughout the meningeal thickness (see Figure 5 in Blaeser et al. 2022). However, the majority of afferents were localized to the most superficial 20 microns (Figure S1E in Blaeser et al. 2022), suggesting that we mostly recorded the activity of dural afferents; (2) to enable simultaneous quantification of three-dimensional deformation and the activity of afferents throughout the thickness of the meninges. This allowed us to determine whether changes in mechanosensitivity could involve augmented activity to intracranial mechanical forces that produced meningeal deformation along the Z-axis of the meninges (e.g., increased intracranial pressure); (3) to provide a direct means to confirm that the afferent GCaMP fluorescent changes we observed were not due to artifacts related to meningeal motion along the Z-axis. We have now added this information to the “Two-photon imaging” section of the Methods.

1. Findings here, from mice with chronic closed cranial windows, failed to fully replicate prior findings from rats with acute open cranial windows. While the species, differing levels of inflammation and intracranial pressure in these two preparations may contribute, as the authors suggested, the modality of measuring neuronal activity could also contribute to the discrepancy. In the present study, conclusions are based entirely on fluorescence signals from GCamp6s, whereas prior rat studies relied upon multiunit recordings/local field potentials from tungsten electrodes inserted in the trigeminal ganglion.As a family, GCamp6 fluorophores are strongly pH dependent, with decreased signal at acidic pH values (at matched Ca2+ concentration). CSD induces an impressive acidosis transient, at least in the brain parenchyma, so one wonders whether the suppression of activity reported in the wake of CSD (Figure 2) in fact reflects decreased sensitivity of the GCamp6 reporter, rather than decreased activity in the fibers. If intracellular pH in trigeminal afferent fibers acidifies in the wake of CSD, GCamp6s fluorescence may underestimate the actual neuronal activity.

Previous in vivo rodent studies observed a tissue acidosis transient that peaks during the DC shift corresponding to the wavefront of the spreading depolarization, and lasting for ~ 10 min. (Mutch and Hansen, 1984). Since we observed a massive increase in afferent calcium activity with a propagation pattern resembling the cortical wave, it is unlikely that the cortical acidosis during the CSD wave strongly affected the GCaMP signal in the overlying meninges. Furthermore, if cortical acidosis non-discriminately affects the GCaMP signal, one would expect a more consistent effect across axons, reflected as a reduced calcium signal in a larger proportion of the afferents, which we did not observe. Finally, the finding that in affected afferents, decreased calcium activity lasted for > 20 min – a time point when cortical acidosis has fully recovered - points to a distinct underlying mechanism. We also note that any residual acidosis would not confound our main finding of increased calcium responses to meningeal deformation at later periods post-CSD, as acidosis should, if anything, decrease calcium-related fluorescence.

The authors might consider injecting an AAV encoding a pHi sensor to the trigeminal ganglion, and evaluating pHi during and after CSD, to assess how much this might be an issue for the interpretation of GCamp6s signals. Alternatively, experiments assessing trigeminal fiber (or nerve/ganglion) activity by electrophysiology or some other orthologous method would strengthen the conclusions.

Please see our comment above regarding the short duration of the pH changes post-CSD.

N's are generally reported as # of afferents, obscuring the number of technical/biological replicates (# of imaging sessions, # of locomotion bouts, # of CSDs induced, # of animals).

We now report the number of replicates (# of afferent, # of CSD events, and # of mice).

Fig 1F trace over the heatmap is not explained in the figure legend. Is this the speed of the running wheel? Is it the apparent propagation rate of the GCamp6s transient through the imaging field?

We have added to the legend of Figure 1 that the trace in panel F depicts locomotion speed.